# The Skeletal Effects of Tanshinones: A Review

**DOI:** 10.3390/molecules26082319

**Published:** 2021-04-16

**Authors:** Sophia Ogechi Ekeuku, Kok-Lun Pang, Kok-Yong Chin

**Affiliations:** Department of Pharmacology, Faculty of Medicine, Universiti Kebangsaan Malaysia, Level 17, Preclinical Building, Jalan Yaacob Latif, Bandar Tun Razak, Cheras Kuala Lumpur 56000, Malaysia; virgosapphire2088@yahoo.com (S.O.E.); pangkoklun@ukm.edu.my (K.-L.P.)

**Keywords:** tanshinones, osteoclastogenesis, osteoblastogenesis, osteoclast, osteoblast, antioxidant, cathepsin inhibitor

## Abstract

Background: Osteoporosis results from excessive bone resorption and reduced bone formation, triggered by sex hormone deficiency, oxidative stress and inflammation. Tanshinones are a class of lipophilic phenanthrene compounds found in the roots of *Salvia miltiorrhiza* with antioxidant and anti-inflammatory activities, which contribute to its anti-osteoporosis effects. This systematic review aims to provide an overview of the skeletal beneficial effects of tanshinones. Methods: A systematic literature search was conducted in January 2021 using Pubmed, Scopus and Web of Science from the inception of these databases. Original studies reporting the effects of tanshinones on bone through cell cultures, animal models and human clinical trials were considered. Results: The literature search found 158 unique articles on this topic, but only 20 articles met the inclusion criteria and were included in this review. The available evidence showed that tanshinones promoted osteoblastogenesis and bone formation while reducing osteoclastogenesis and bone resorption. Conclusions: Tanshinones modulates bone remodelling by inhibiting osteoclastogenesis and osteoblast apoptosis and stimulating osteoblastogenesis. Therefore, it might complement existing strategies to prevent bone loss.

## 1. Introduction

Osteoporosis is a condition that increases the fracture risk of patients due to decreased bone strength, which is a result of bone microstructure and declining bone mass [1,2]. Osteoporosis can occur in both sexes, but it is more common in postmenopausal women than their male counterparts. This sex distinction occurs because of lower peak bone mass in women and the rapid decline of bone mass due to sudden cessation of oestrogen production in the body. Oestrogen deficiency leads to increased bone turnover, subsequently bone loss and osteoporosis [3]. Osteoporosis frequently remains undiagnosed due to its asymptomatic nature until it presents as low-trauma hip, spine, proximal humerus, pelvis and/or wrist fractures [4,5]. In 2010, it was estimated that 158 million individuals globally, aged 50 years or older, were at high risk of osteoporotic fracture and this number is expected to double by 2040 [6]. The cost of fractures in the United States is estimated to reach $25 billion annually by 2025 corresponding to three million projected fractures [7].

Pharmacotherapeutics for osteoporosis could be divided into anti-resorptive (i.e., bisphosphonates, oestrogen receptors modulators, oestrogen replacement and denosumab) or anabolic (i.e., teriparatide) medications. Anti-resorptive osteoporotic drugs reduce the rate of bone resorption by inhibiting osteoclasts [8]. Although they manage osteoporosis effectively, bone resorption inhibition is often associated with reduced bone formation since both processes are coupled. Coupling is modulated by osteogenic variables produced by osteoblasts. This event will prevent the repair of bone micro-fractures and jeopardise skeletal microarchitecture [9,10]. Cathepsin K (CatK) inhibitors allow inhibition of bone resorption without disturbing bone formation [11], but most agents are still in development [12,13,14]. Odonacatib is the only CatK inhibitor that had entered phase III clinical trial but was terminated due to cardio-cerebrovascular adverse effects [15,16]. On the other hand, anabolic agents increase bone formation rate more than bone resorption [8], but they are offered to patients with high fracture risk and have failed other therapies [11]. Other preventive agents like calcium and vitamin D are available, but their effectiveness is inconsistent. Ensuring adequate dietary calcium and vitamin D is compulsory in stopping the progression of osteoporosis [17]. However, vitamin D and calcium supplementation alone might not be sufficient to stop osteoporosis from occurring [18].

Tanshinones are the lipid-soluble diterpenoid quinones isolated from Danshen, the dried roots of a well-known traditional medicine *Salva miltiorrhiza*. They are the major lipid-soluble pharmacological constituents of Danshen and give the root its reddish-brown colour [19]. The major tanshinones isolated from Danshen included 15,16-dihydrotanshinone (D-T), tanshinone I (T-I), cryptotanshinone (C-T) and tanshinone IIA (T-IIA) [20,21]. A previous study indicated that D-T could selectively block the collagenase activity of CatK without affecting protease activity and osteoclastogenesis [22]. In vitro studies have also reported that T-IIA and C-T possess anti-inflammatory effect by inhibiting the activation of nuclear factor kappa B (NF-κB) pathway [23,24,25,26]. These studies suggest the potential of tanshinones as an anti-osteoporotic agent. Therefore, this article aims to review the effects of tanshinones on bone based on evidence from in vitro and in vivo studies.

## 2. Results

### 2.1. Selection of Articles

A total of 158 articles were retrieved from the literature search, of which 59 were from PubMed, 89 were from Scopus and 10 were Web of Science. After removing duplicates (*n* = 61), 97 articles were screened. Seventy-seven articles not meeting the selection criteria were eliminated (12 review articles, 1 editorial letter and 64 articles not relevant to the current review). Finally, 20 articles fulfilling all criteria mentioned were included in the review (Appendix A). The selection process from identification, screening, eligibility to inclusion of articles is shown in Figure 1.

### 2.2. Study Characteristics

The selected studies were published from 2004 to 2020. Thirteen studies were in vitro experiments using primary cells from mouse (bone marrow cells (mBMCs), calvarial osteoblasts), human (bone marrow cells (hBMCs), periodontal ligament stem cells (hPDLSC), bone marrow mononuclear cells (BMMCs), bone marrow mesenchymal stromal cells (BM-MSCs)) and cell lines (RAW264.7, MC3T3-E1, C2C12) [27,28,29,30,31,32,33,34,35,36,37,38,39]. Twelve in vivo studies using Sprague Dawley rats, Wnt1^sw/sw^ mice and C57BL/6J mice were included. Kunming (KM) mice and ICR mice [30,32,36,37,39,40,41,42,43,44,45,46]. No human studies on this topic were reported.

The in vitro studies investigated the effects tanshinone or its derivatives on osteoclast differentiation using macrophage colony-stimulating factor (M-CSF), receptor activator of NF-ĸB (RANK) ligand (RANKL), interleukin 1 alpha (IL-1α), lipopolysaccharide (LPS) or tumour necrosis factor-alpha (TNF-α). They also studied the osteogenic effects and anti-apoptotic effects of tanshinone in the presence of dexamethasone. The doses of tanshinone or its derivatives used ranged between 0.001 and 1000 µM and 1.5 mg/mL (5428.88 µM) [37]. The treatment period was 5–7 days for the differentiation of osteoclasts and 1–7 days for the differentiation of osteoblasts.

For animal studies, the doses of tanshinone used were between 0.36 and 200 mg/kg [30,32,36,37,39,40,41,42,43,44,45,46]. The disease model used in these studies included retinoic acid, ovariectomy (OVX)/oestrogen deficiency, streptozocin/diabetes, lipopolysaccharide/inflammation and collagen/rheumatoid arthritis-induced bone loss, as well as polyethylene (PE) particle-induced calvarial osteolysis mimicking osteolysis in arthroplasty. These studies determined bone microstructure using micro-computed tomography, bone histomorphometry and bone remodelling markers to assess bone health. Table 1 summarises the effects of tanshinones on bone health.

### 2.3. Results from Cell Culture Studies

Osteoblasts are specialised mesenchymal cells that synthesise bone matrix and coordinate the bone mineralisation [47]. Liu et al. reported that 48 h incubation with tanshinone IIA (T-IIA) (2.5 and 5 μM) increased osteogenic differentiation in human periodontal ligament stem cells (hPDLSC) compared to the control group [27]. Gene expression studies showed that T-IIA upregulated osteoblastogenesis-related genes like osteocalcin (OCN), osteopontin (OPN), runt-related transcription factor 2 (Runx2), alkaline phosphate (ALP). Pre-treatment with T-IIA (1, 5, 10 and 20 μM for 7 days) on BM-MSCs increased osteoblast number at early osteogenesis stages as evidenced by increased ALP expression [28]. Besides this, gene expression studies showed decreased receptor activator of nuclear factor-κΒ ligand (RANKL) expression and increased OPN, osteoprotegerin (OPG), Runx2, bone morphogenetic protein (BMP)-2, ALP and collagen 1 expression [28]. C2C12 cells are muscle cell lines that can differentiate into osteoblasts when cultured in monolayers under the influence of BMP2 [48]. Kim and Kim reported T-IIA (2.5, 5, 10 and 20 μM) for 7 days increased BMP-2-induced trans-differentiation of C2C12 cells into osteoblasts [35]. T-IIA treatment also increased activation of osteogenic genes like ALP, OCN, Runx2, BMP-2, 4, 6, 7 and 9. Similarly, Wang et al. [36] reported that T-IIA (5, 10, 15 and 20 μM for 48 h) increased MC3T3-E1 pre-osteoblast differentiation by increasing Runx2, osterix expression and ALP activity. T-IIA treatment also increased BMP-2 protein expression and activated the c-Jun N-terminal kinases (JNK) signalling pathway [36].

Besides this, T-IIA pre-treatment (1 μM) on MC3T3-E1 pre-osteoblasts for 24 h reduced dexamethasone-induced osteoblast apoptosis by inhibiting NADPH oxidase 4 (Nox4) expression [38]. Nox4 is an enzyme involved in reactive oxygen species (ROS) production [49], and ROS has been reported to promote osteoblast apoptosis [50]. Zhu et al. reported that T-IIA treatment (1.5 mg/mL for 24 h) increased osteoblast viability and decreased osteoblast apoptosis in osteoblast cells by increasing the expression of anti-apoptotic genes such as p53 and B-cell lymphoma 2 and reducing expression of apoptotic genes like apoptotic protease-activating factor 1 and caspase-3 in osteoblasts [37]. T-IIA also exhibited protective effect against oxidative stress in hydrogen peroxide-treated osteoblasts by reducing ROS, thiobarbituric acid reactive substance (TBARS) and reactive nitrogen species (RNS) but did not improve superoxide dismutase level or increase ALP expression [37]. T-IIA could improve osteoblast survival by regulating the NF-κB signalling pathway. In the study, NF-κB phosphorylation in osteoblasts was decreased, the expression levels of NF-κB targets genes, such as TNF-α, inducible nitric oxide synthase (iNOS) and COX-2 and proteins such as p65, an inhibitor of NF-κB alpha (IκBα) and IκB kinase beta (IKK-β) were reduced with T-IIA treatment. However, the expression of tumour necrosis factor receptor-associated factor (TRAF)-1, an inhibitor of apoptosis protein, was increased in the T-IIA-treated osteoblasts [37].

On the other hand, T-IIA treatment (1, 2 and 5 μg/mL) for 7 days reduced the number of tartrate-resistant acid protein (TRAP) positive cells formed from RANKL-treated RAW 264.7 cells in a dose-dependent manner than the untreated cells [39]. Osteoclast function was also reduced dose-dependently as evidenced by decreased resorbing pits and actin rings. Gene expression studies showed that T-IIA reduced osteoclastogenesis by suppressing osteoclastogenesis-related genes, such as nuclear factor of activated T-cells cytoplasmic 1 (NFATc1), TRAP, matrix metalloproteinase 9, cathepsin K, calcitonin receptor, and TRAF 6 [39]. T-IIA also suppressed osteoclastogenesis by inhibiting RANKL-induced activation of NF-κB, as evidenced by inhibition of p65 translocation, mitogen-activated protein kinase (MAPK) and protein kinase B (Akt) pathways [39]. Similarly, Kim et al. reported that pre-treatment with T-IIA (2.5, 5, 10 and 20 μg/mL) for 1 h reduced osteoclast fusion, actin ring formation and resorption area in macrophage colony-stimulating factor (M-CSF) and RANKL-treated calvarial osteoblasts and BMCs coculture [33]. T-IIA inhibited osteoclast differentiation marked by reduced calcitonin receptor, c-Src and integrin β3 expression [33]. Pre-treatment with T-IIA also inhibited the activation of ERK, Akt and NF-κB signal transduction pathways [33]. Kwak et al. [31] reported a reduction in osteoclast differentiation, c-Fos and NFATc1 expression following T-IIA treatment (10 μg/mL) for 4 days in RANKL and M-CSF-treated BMCs and calvarial osteoblast coculture. The anti-osteoclastogenic activity of T-IIA was mainly exerted through its anti-inflammatory properties. Mechanistically, T-IIA reduced the RANKL-induced activation of NF-κB by blocking the p65 subunit nuclear translocation [33,39]. T-IIA reversed the RANKL-induced activation of MAPK and Akt pathways by reducing the levels of ERK, JNK, c-FOS and Akt expression [31,33,39]. Kwak et al. [32] reported that T-IIA treatment (5 μg/mL) for 6 days inhibited osteoclast differentiation in cocultures of calvarial osteoblast cells and BMCs treated with tumour necrosis factor-alpha (TNF-α), interleukin 1 alpha (IL-1α) or LPS. T-IIA regulated the expression of RANKL and OPG in osteoclasts treated with LPS. A mechanistic study revealed that T-IIA reduced osteoclast formation and regulated expression of RANKL and OPG by inhibiting prostaglandin E2 (PGE_2_) through selective COX-2 inhibition [32].

Tanshinone II A sulfonic sodium (T06) is a water-soluble derivative of T-IIA [51] developed to increase the bioavailability of T-IIA [52]. Panwar et al. reported that treatment with T06 (200 and 500 nM) for 72 h effectively reduced cross-linked C-telopeptide of type I collagen (CTX-1) level in the culture supernatant of M-CSF and RANKL-treated human and mouse BMCs cultured on bone slides [30]. Treatment with 1 μM of T06 for 72 h reduced the total eroded surface of dentin slides created by mouse and human osteoclasts. However, it did not affect the metabolic activity and number of osteoclasts, suggesting that it was not toxic to the cells [30]. The main difference between T-IIA, T-I, D-T and C-T is in their structure and IC_50_ values. T-IIA and C-T have a dimethyl tetranaphthalene ring with IC_50_ values of 89.1 and 226.7 µM, respectively, while T-I and D-T contain naphthalene rings A and B with IC_50_ values of 38.7 and 14.4 µM respectively [53]. Lee et al. reported that T-IIA, T-1, D-T and C-T (0.5, 1 and 2.5 μg/mL for 3 days) decreased TRAP-positive multinuclear cells in M-CSF treated primary osteoblasts and bone marrow cell coculture [29]. Similarly, Kim et al. reported that T-IIA, T-I, D-T and C-T (1 μg/mL for 7 days) reduced osteoclastogenesis, evidenced by a reduced number of TRAP-positive cells but did not affect osteoblastogenesis due to the lack of change in ALP activity in RANKL and M-CSF-treated MC3T3-E1 cells [34].

### 2.4. Results from Animal Studies

T-IIA is the most widely studied tanshinone type, shown to produce skeletal beneficial effects in both healthy and animals suffering from bone loss. T-IIA intake (22 mg/kg/day for 8 weeks) significantly increased the whole-body and femoral BMD, maximum femoral load and histomorphometric indices of healthy female Wistar rats. Increased serum OCN and lower TRAP levels were also observed in the treated rats [46].

On the other hand, postmenopausal osteoporosis is primarily caused by oestrogen deficiency, which disrupts normal bone remodelling by increasing osteoclastic resorption activity more than bone formation, leading to a net loss of bone [54]. Wang et al. reported that T-IIA treatment (10 mg/kg for 2 weeks) increased bone volume, trabecular number (Tb.N), trabecular thickness (Tb.Th) and reduced trabecular separation (Tb.Sp) in Sprague Dawley rats with OVX-induced osteoporosis [40]. Similarly, OVX osteoporotic C57BL/6 mice treated with T-IIA (10 mg/kg for 6 weeks) showed reduced trabecular bone loss [39]. They also showed increased bone surface area/total volume, BV/TV, BMD, Tb.N and decreased trabecular pattern factor [39]. Cui et al. [43] reported that treatment with 200 mg/kg of total tanshinone (equivalent to 35 mg/kg of T-IIA + 16 mg/kg of C-T) for 30 weeks increased BV/TV, Tb.Th and reduced osteocalcin/bone surface, mineralisation rate and bone formation rate at the proximal tibial metaphysis of OVX osteoporotic rats. Additionally, T-IIA significantly restored the cortical bone thickness and Tb.N with higher serum estradiol levels and lower serum phosphate, ALP and TRAP levels than untreated KM mice with retinoic acid-induced osteoporosis [44]. Panwar at al. [30] reported that treatment with T06 (40 mg/kg for 3 months) could improve the structural properties of OVX C57BL/6 mice by reducing Tb.Sp and increasing BV/TV and Tb.N. Immunohistochemical analysis revealed that T06 treatment increased the number of osteoblasts per bone perimeter, confirmed by increased ALP expression in Western blot analysis and increased plasma procollagen-1 *N*-terminal peptide concentration [30]. However, T06 treatment did not affect total osteoclast numbers/bone surface and osteoclast numbers/bone perimeter than the OVX mice [30].

A spontaneous mutation in *WNT1* (one of the major WNT ligands regulating bone homeostasis) is present in the swaying (Wnt1^sw/sw^) mouse, wherein a recessive mutation causes osteogenesis imperfecta and heterozygous mutation causes the early onset of osteoporosis [55]. In Wnt1^sw/sw^ osteoporotic mice, T-IIA treatment (10 mg/kg for 6 weeks) increased the stiffness, ultimate strength and elastic modulus compared to mice treated with alendronate [37]. Diabetes has a major impact on bone stability, since both type 1 and type 2 diabetes elevate fracture risk, making skeletal fragility a complication of diabetes [56,57]. In 8-week-old C57BL/J6 mice with streptozotocin-induced diabetic osteoporosis, T-IIA treatment (10 and 30 mg/kg; 3 times a week for 8 weeks) increased the bone mineral density (BMD) of trabecular bone compared to the negative control [42]. Improvement in the bone structural properties was marked by increased volumetric BMD, BV/TV, bone area fraction, connectivity density and a decrease in the structural model index (SMI) [42]. Inflammation also directly affects bone remodelling because proinflammatory cytokines have been implicated in the regulation of osteoblasts and osteoclasts [58]. Kwak et al. [32] reported that ICR mice with LPS-induced bone loss showed decreased osteoclast formation and cancellous bone loss when treated with T-IIA (5 μg/g on days 1, 3, 5 and 7).

Yao et al. reported that T-IIA supplementation (1 and 2 μg /g/day for 21 days) inhibited bone resorption by reducing the number of resorption pits, percentage of porosity in the skull and TRAP-positive cells in 8 weeks old C57BL/J6 mice with PE particle-induced calvarial osteolysis. These events increased BV/TV, BMD and bone strength of the rats [41]. In this study, T-IIA achieved its bone protective potential by decreasing osteoclastogenic markers, such as RANKL/OPG ratio, osteoclast-associated immunoglobulin-like receptor and CTX-1 expression [41]. Zhang et al. also demonstrated that direct gingival mucosa injection of T-IIA significantly reduced the recurrence distance and percentage of first molar tooth movement on Wistar rats by suppressing the osteoclast activity as evidenced by the higher ratio of OPG to osteoclast differentiation factor [45]. Wang et al. also observed that T-IIA treatment (10, 20 and 30 mg/kg for 4 weeks) promoted fracture healing in mice that underwent femur diaphysis osteotomy by enhancing the volume of callus area after damage, callus intensity, low-density bone volume/callus total volume, tissue mineral density and BMD [36].

## 3. Materials and Methods

### 3.1. Literature Review

This systematic review was performed in line with Preferred Reporting Items for Systematic Reviews and Meta-Analyses (PRISMA) (Appendix A) A systematic literature search was conducted in January 2021 using PubMed, Scopus and Web of Science database to identify studies on the anti-osteoporotic properties of tanshinones. The search string used was (1) tanshinones AND (2) (bone OR osteoporosis OR osteoblasts OR osteoclasts OR osteocytes).

### 3.2. Article Selection

Articles with the following characteristics were included: (1) original research articles with the primary objective of determining the anti-osteoporotic effects of tanshinones; (2) studies using cell cultures, animal models or human subjects. Articles were excluded if they were: (1) conference abstract, review, letter/commentary, editorial, perspective articles without original data; (2) studies using extracts/formulations containing tanshinones and other bioactive compounds; (3) articles not written in English. Two authors performed the search using both databases and search string mentioned. Inclusion of an article in the review was based on the consensus of both authors. If a consensus could not be reached, the corresponding author would decide on the fate of the article.

### 3.3. Data Extraction

Data extracted included authors’ names, publication year, study design, dose and treatment period, findings and limitations of the study.

## 4. Discussion

The current literature revealed that tanshinones exhibited anti-osteoporotic activity by suppressing osteoclastogenesis and improving osteoblastogenesis. Tanshinones exert anti-osteoclastogenic activity by blocking RANKL-induced activation of NF-κB, MAPK, Akt pathways and M-CSF/c-Src signalling. Tanshinones prevent osteoblast apoptosis and facilitate osteoblastogenesis by ameliorating oxidative stress and inflammation. Animal studies revealed that tanshinones prevented skeletal deterioration in osteoporosis models induced by oestrogen deficiency, diabetes and inflammation. They also prevent PE-induced osteolysis and improve fracture healing in animal models.

Osteoblasts are mesenchymal cells that produce and mineralise the bone matrix [47]. Osteoblast differentiation is regulated by Runx2 phosphorylation and transcription, which is mediated by the MAPK cascades. MAPK pathways and its components, JNK, ERK and p38, form the non-canonical BMP2 signal transduction pathways that regulate osteoblastogenesis [59,60,61,62]. T-IIA increases the activation of osteogenic genes which indicate its stimulatory effect on osteogenic differentiation [27,28,35]. Wang et al. pinpointed that the effects of T-IIA were mediated through JNK as inhibition of this pathway negated its effects on osteoblast differentiation and mineralisation [36].

Excessive ROS production could overwhelm intracellular antioxidant defence, causing oxidative stress and osteoporosis [63,64]. Apart from inhibiting osteoblast proliferation [65] and differentiation [66], oxidative stress also induces osteoblast apoptosis [67,68], thereby jeopardising bone formation [69]. Ameliorating oxidative stress could decrease osteoblast apoptosis and increase its differentiation [37]. Hydrogen peroxide, an end-product of Nox4, is one of the major ROS predominately present in mitochondria [70,71]. Blocking Nox4 activation may prevent osteoblast apoptosis. T-IIA was reported to inhibit Nox4 expression, which in turn decreased osteoblast apoptosis [38]. NF-κB activation negatively regulates bone formation by suppressing osteoblast differentiation [72]. T-IIA increased osteoblast differentiation via suppressing NF-κB activation and translation of its target genes (TNF-α, iNOS and COX2). This event is achieved by preventing IKK-β and IκBα degradation and p65 nuclear translocation. These effects translated to improved bone microstructure and biomechanical properties in animals treated with T-IIA [37].

Osteoclasts are derived from hematopoietic lineage cells and are capable of bone resorption [73]. It is necessary for normal bone homeostasis, but excessive resorption can induce pathological bone loss. A variety of hormones and cytokines regulate osteoclast differentiation and activation. Particularly, RANKL and M-CSF are the essential cytokines for osteoclastic differentiation [74]. M-CSF binds to colony-stimulating factor 1 receptor, and RANKL binds to RANK to promote osteoclast differentiation and survival, as well as bone resorptive activity [75,76]. RANK-RANKL binding leads to the recruitment of TRAF factors like TRAF 6 [77], resulting in the activation of NF-kB, Akt and MAPKs (ERK/p38/JNK) pathways. Besides, RANKL signalling activates c-Fos and subsequently NFATc1, a master switch for controlling osteoclast terminal differentiation [78,79,80]. All these pathways act in concert to initiate osteoclast differentiation and bone resorption by inducing transcription and expression of osteoclast-specific genes, such as TRAP, cathepsin K, matrix metalloproteinase 9 (MMP-9) and C-Src [81]. Exogenous factors like LPS can alter RANKL signalling and influence osteoclastogenesis. LPS induces production of proinflammatory cytokines of osteoblasts and precursor cells via COX-2, especially TNF-α, which subsequently augment RANKL signalling in osteoclasts [82,83]. Blocking of COX-2 is reported to inhibit osteoclastogenesis in vitro [84,85].

T-IIA treatment is reported to suppress RANKL and M-CSF-induced osteoclastogenesis from precursor cells and osteoclast function [33,39]. Similar effects were observed with T06, T-1, C-T and D-T treatment, indicating all tanshinones share similar properties [29,30,34]. T-IIA prevents RANKL-induced activation of TRAF 6, which in turn reduces activation of NF-κB, MAPK, Akt and c-Src pathways, and inhibits osteoclast formation and activity, marked by reduced TRAP and MMP9 expression [33,39]. T-IIA also inhibits the expression of RANKL-induced c-Fos and NFATc1, which suppress osteoclast differentiation [31]. Besides, T-IIA is reported to inhibit LPS-mediated COX-2 expression in bone marrow and calvarial osteoblast cells [32]. This action could reduce osteoclast formation and bone loss prevention in mice administered with LPS [32].

RANK-RANKL signalling also regulates CatK expression [86]. CatK represents a potential target of anti-osteoporosis therapy. CatK inhibition does not affect bone formation [87,88], suggesting that bone formation and resorption are uncoupled [16]. Furthermore, osteoclast formation and survival required for osteoblastic bone formation response during remodelling are not affected by CatK inhibition [89]. In contrast, increased bone formation was observed in CatK deficient mice [90]. The different forms of tanshinones (T-IIA and T06) exert distinct effects in CatK-associated bone remodelling. T-IIA suppressed RANKL-induced expression of CatK in RAW264.7 cells and BMMCs, but reduced the function and survival of osteoclasts [39]. This observation suggests that T-IIA may not be a selective inhibitor of CatK. On the other hand, T06 inhibited the activity of CatK but did not alter CatK-positive osteoclast number in mice with OVX-induced osteoporosis [30], which suggests that T06 may be a selective inhibitor of CatK. The selectivity ensures normal osteoblast-osteoclast crosstalk and bone remodelling are not interrupted. In comparison, bisphosphonates, an antiresorptive agent commonly used as the first-line treatment for osteoporosis, also inhibit bone formation [91].

The biological effects of tanshinones suggest their various potential clinical applications. Direct gingival mucosa injection of T-IIA reduced the recurrence distance and percentage of first molar tooth movement in Wistar rats by suppressing the osteoclast activity [45]. This evidence showed that T-IIA could prevent the loosening of teeth in gingival and periodontal diseases. Tanshinones enhanced skeletal health in healthy animals, by increasing BMD, femoral microstructures and strength, as well as lowering osteoclast activity [46]. Therefore, it could serve as an agent for the primary prevention of osteoporosis. Furthermore, various skeletal actions of tanshinones could be harnessed for secondary prevention of osteoporosis. T-IIA restored retinoic acid-induced decrease of cortical bone thickness and Tb.N by increasing serum oestradiol levels and preventing high bone remodelling in Wistar rats [44]. T-IIA also improved bone structural properties in mice with STZ-induced diabetic osteoporosis [42]. Additionally, T-IIA [39,40], T06 [30] and total tanshinones (T-IIA + C-T) [43] prevented OVX-induced bone loss in rats and mice by improving bone microstructures. T06 increased osteoblast number [30] while total tanshinones decreased osteoclast number [43] in vivo. Untreated osteoporosis could lead to fragility fracture. T-IIA could increase fracture healing in mice with femoral osteotomy to mimic fracture [36]. Osteolysis following joint replacement is mainly caused by the abrasive particles introduced by the prosthesis [41]. Recent studies showed that these abrasive particles could induce the release of cytokines associated with osteolysis, such as IL-6, IL-1, TNF-α and PGE_2_, worsening the inflammatory response [92]. PE particles have been confirmed to induce osteolysis around artificial joints [93]. T-IIA was shown to prevent calvarial bone resorption in PE particle-induced osteolysis [41]. Hence, T-IIA could be embedded with arthroplasty materials to avoid triggering inflammation and bone resorption.

Limited studies examined the safety of tanshinones. T-IIA at high concentration (≥6 μM) caused severe growth inhibition, development malformation and cardiotoxicity in zebrafish [94]. Similarly, T-11A at a high concentration (25 μM) was toxic to human endothelial EAhy926 cells as these cells were killed after a 24 h treatment [95]. There is limited data on the pharmacokinetic properties of tanshinones. Zhang et al. reported that tanshinones have limited bioavailability when administered orally [96]. After oral administration at 100 mg/kg, the systemic bioavailability of C-T was 2.05%, suggesting poor absorption or significant metabolism in the gut or/and liver. After intraperitoneal administration at 100 mg/kg, the systemic bioavailability of C-T was 10.60%, suggesting hepatic metabolism or low solubility of C-T [97]. There is a paucity of pharmacokinetics, pharmacodynamics and safety data of tanshinones in humans. A search through https://clinicaltrials.gov/ (accessed on 30 March 2021) using the term “tashinones” revealed four registered trials on left ventricular remodelling secondary to acute myocardial infarction (identifier: NCT02524964), pulmonary hypertension (identifier: NCT01637675), polycystic ovary syndrome (identifier: NCT01452477) and childhood acute promyeloid leukemia (identifier: NCT02200978). The last trial was recruiting subjects, while the status of the other three trials was unknown. No human clinical trial on the effects of tanshinones on skeletal diseases has been attempted, probably due to poor intestinal absorption and bioavailability [97,98]. Various methods have been developed to address the problem of low bioavailability, including designing water-soluble tanshinone derivatives, loading of tanshinones into discoidal and spherical high-density lipoproteins, liposomes, nanoparticles, microemulsions, cyclodextrin and solid dispersions [99,100,101,102,103]. However, due to the complicated manufacturing process, little progress has been made in the clinical application of these formulations [104]. Therefore, more comprehensive studies in this regard will help to establish tanshinones as one of the clinical therapeutic options for various bone conditions.

The present review did not exclude studies based on quality and assumed that articles published by journals indexed in three main and reputable databases are of sufficient quality to be included. We did not consider articles published in non-indexed journals, unpublished articles, thesis and non-primary research articles, potentially excluding some relevant studies.

## 5. Conclusions

The preclinical evidence so far supports that tanshinones exert skeletal protective effects by inhibiting osteoclastogenesis/bone resorption and promotes osteoblastogenesis/bone formation. A type of tanshinones, T06, is shown to be a specific CatK inhibitor, which reduces bone degradation without affecting the crosstalk between osteoblasts and osteoclasts. There is a lack of human clinical trials to validate the skeletal effects of tanshinones. Thus, well-planned clinical trials should be conducted to endorse the skeletal properties of tanshinones and prove the safety of these compounds.

## Figures and Tables

**Figure 1 molecules-26-02319-f001:**
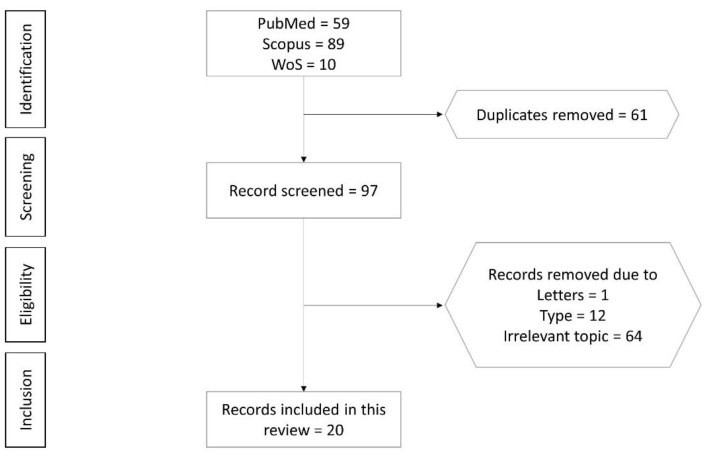
Flowchart of the article selection process.

**Table 1 molecules-26-02319-t001:** Effects of tanshinone on bone health.

Studies	Study Design	Changes with Tanshinone Treatment
Cell Culture Studies		
Liu et al. [27]	Cell: hPDLSC from the premolars of 20 donors without oral or systematic diseases (10 men and 10 women aged 12–25 years old) Model: osteogenic differentiation Treatment: 2.5 and 5 μM of T-IIA for 48 h Negative control: Untreated cells Positive control: N.A.	↑ osteogenic differentiation ↑ OCN, OPN, Runx2, ALP gene expression vs. negative control
Qian et al. [28]	Cell: BM-MSCs isolated from the tibia and femur of BALB/cJ mice (4–6 weeks old) Model: dexamethasone-induced osteogenic differentiation Treatment: 1, 5, 10 and 20 μM of T-IIA for 7 days Negative control: Untreated cells Positive control: N.A.	↑ ALP expression vs. negative control ↑ OPN, OPG, collagen 1 and ↓ RANKL vs. negative control ↑ Runx2 and BMP 4 vs. negative control
Kim and Kim [35]	Cell: C2C12 cells Model: BMP-2-induced osteoblast differentiation Treatment: 2.5, 5, 10 and 20 μM of T-IIA for 7 days Negative control: Untreated cells Positive control: N.A.	↑ BMP-2-induced osteoblast differentiation and ALP production ↑ activation of osteogenic genes (ALP, OCN, Runx2, BMP-2, 4, 6, 7, 9)
Wang et al. [36]	Cell: MC3T3-E1 cells Model: osteoblast differentiation Treatment: 5, 10, 15 and 20 μM of T-IIA for 48 h Negative control: Untreated cells Positive control: N.A.	↑ Runx2, Osx expression and ALP activity vs. negative control ↑ BMP-2 protein expression vs. negative control ↑ activation of JNK pathway vs. negative control
Li et al. [38]	Cell: MC3T3-E1 cells Model: dexamethasone-induced osteoblast apoptosis Treatment: 1 μM of T-IIA for 24 h Negative control: Untreated cells Positive control: N.A.	↓ dexamethasone-induced osteoblast apoptosis by inhibiting Nox4 expression
Zhu et al. [37]	Cell: Osteoblast cells from 10 weeks old female Wnt1^sw/sw^ mice Model: H_2_O_2_-induced osteoblast apoptosis Treatment: 1.5 mg/mL of T-IIA for 24 h Negative control: PBS-treated cells Positive control: 1.5 mg/kg alendronate	↑ osteoblasts viability and ↓ osteoblast apoptosis ↓ ALP, H_2_O_2_, ROS, SOD, TBARS and RNS levels vs. osteoblasts from mice treated with alendronate ↓ caspace-3, and Apaf-1 expression and ↑, Bcl-2, TRAF 1, IAP and p53 expression vs. osteoblasts from mice treated with alendronate ↓ activation of NF-κB phosphorylation, NF-κB activity, TNF-α, iNOS and COX2 expression vs. osteoblasts from mice treated with alendronate ↓ p65, IKK-β and IκBα vs. osteoblasts from mice treated with alendronate
Cheng et al. [39]	Cell: RAW264.7 cells and BMMCs isolated from the femoral bone marrow of 8-week-old C57BL/6 mice. Model: RANKL-induced osteoclastogenesis Treatment: 1, 2 or 5 μg/mL of T-IIA for 7 days Negative control: Untreated cells Positive control: N.A.	↓ osteoclastogenesis and osteoclast function vs. negative control ↓ TRAP+ cells (in a dose-dependent manner) vs. negative control ↓ numbers and size of actin ring structures (a characteristic feature of mature osteoclasts during osteoclastogenesis) vs. the negative control ↓ RANKL-induced osteoclast differentiation at early stages vs. negative control ↓ osteoclastogenesis-related genes (TRAP, MMP- 9, cathepsin K, CTR, and TRAF6) vs. negative control ↓ RANKL-induced activation of the NF-κB, MAPK and Akt pathways ↓ ERK, JNK, c-Fos and Akt expression in osteoclasts
Lee et al. [29]	Cell: Primary osteoblasts from calvarial cells of ICR newborn mice/bone marrow cells obtained from tibiae of 6- to 7-week-old ICR mice (coculture) Model: M-CSF-induced osteoclast differentiation Treatment: 0.5, 1 and 2.5 μg/mL of T-IIA, T-I, C-T, D-T for 3 days (osteoblasts) and 6-7 days (osteoclasts) Negative control: Untreated cells Positive control: N.A.	↓ TRAP-positive multinuclear cells vs. negative control ↓ viability of bone marrow cells following treatment with T-I, C-T, D-T NS in the viability of bone marrow cells following treatment with T-IIA
Panwar et al. [30]	Cell: mononuclear cells from human bone marrow and bone marrow cells from femur and tibia from 4 months old mice Model: M-CSF and RANKL-induced osteoclastogenesis Treatment: 1 and 3 μM of T06 for 72 h Negative control: DMSO (1%)-treated cells Positive control: N.A.	↓TRAP-stained osteoclasts and toluidine-stained resorption events in mouse and human osteoclasts vs. negative control ↓ total eroded surface in human and mouse osteoclasts vs. negative control ↓ CTx-1 expression vs. negative control NS for osteoclastogenesis in bone marrow mononuclear cells
Kim et al. [34]	Cell: MC3T3-E1 cells and bone marrow cells isolated from the long bone of 7-weeks-old ICR male mice Model: M-CSF and RANKL-induced osteoclastogenesis Treatment: 1 μg/mL of T-IIA, T-I, D-T, and C-T for 7 days Negative control: DMSO-treated cells Positive control: N.A.	↓ osteoclastogenesis in all tanshinone isoforms vs. negative control NS for osteoblastogenesis in all tanshinone isoforms vs. negative control
Kwak et al. [31]	Cell: Bone marrow cells from tibia and femur and mouse calvariae from pericranium of 5 weeks old male ICR mice. Model: M-CSF and RANKL-induced osteoclastogenesis Treatment: 10 μg/mL of T-IIA for 4 days Negative control: M-CSF-treated cells Positive control: N.A.	↓ RANKL-mediated osteoclast differentiation vs. negative control ↓ c-Fos and NFATc1 expression induced by RANKL
Kim et al. [33]	Cell: Calvarial osteoblasts from the new bone of ICR mice and bone marrow cells from tibiae of 6–7 weeks old ICR mice (Mouse bone marrow cells and calvarial osteoblast coculture) Model: M-CSF and RANKL-induced osteoclastogenesis Treatment: 2.5–20 μg/mL of T-IIA for 15 min–20 h Negative control: Untreated cells Positive control: N.A.	↓ osteoclast differentiation, osteoclast fusion, actin ring formation and resorption area vs. negative control ↓ osteoclast differentiation-related genes (calcitonin receptor, c-Src, and integrin β3) ↓ activation of ERK, Akt and NF-κB signal transduction pathways
Kwak et al. [32]	Cell: Calvarial osteoblasts and bone marrow cells isolated from the femur and tibias of 5-weeks-old ICR male mice (Mouse bone marrow cells and calvarial osteoblast coculture) Model: TNF-α, IL-1α or LPS-induced osteoclast differentiation Treatment: 5 μg/mL of T-IIA for 7 days Negative control: Untreated cells Positive control: N.A.	↓ osteoclast differentiation vs. negative control ↓ LPS-induced RANKL and OPG expression in osteoblasts vs. negative control ↓ LPS-mediated COX-2 expression and LPS-induced PGE_2_ in osteoblasts
**Animal studies**		
Zhang, et al. [45]	Animals: 48 male Wistar rats (2 months old) Experimental model: Relapse stage after orthodontic mesial movement of maxillary first molar tooth Treatment: 0.36, 0.72 and 1.44 mg/day of T-IIA for 4 weeks (localised gingival mucosa injection) Negative control: Normal saline injection Positive control: N.A.	↓ recurrence distance and percentage of tooth movement by regulating osteoclast activity with ↑ OPG/osteoclast differentiation factor NS in body weight changes
Yang, et al. [46]	Animals: 40 female Wistar rats (1 month old, 97 ± 3 g Disease model: N.A. Treatment: 22 mg/kg/day of T-IIA for 8 weeks (oral) Negative control: untreated rats Positive control: N.A. Comparative groups: 16.8 mg/kg/day of resveratrol and T-IIA (11 mg/kg/day) + resveratrol (8.4 mg/kg/day) for 8 weeks	↑ serum OCN, whole-body and femoral BMD, maximum femoral load and histomorphometric indices (trabecular width, trabecular separation degree, Tb.N and trabecular area) vs. untreated rats ↓ serum TRAP vs. untreated rats NS in body weight
Wang et al. [36]	Animals: 40 male C57BL/J6 mice (3 months old) Disease model: Open osteotomy at femur diaphysis Treatment: 10, 20 and 30 mg/kg/day of T-IIA for 4 weeks (oral) Negative control: Mice with open osteotomy received with methanol Normal control: No treatment or surgery Positive control: N.A.	↑ callus area, callus intensity, BV1/TV, TMD and BMD vs. negative control
Yao et al. [41]	Animals: 24 male C57BL/J6 mice (2 months old) Disease model: PE particle-induced calvarial osteolysis Treatment: 1 and 2 μg/g/day of T-IIA for 21 days (periosteum injection) Negative control: PE-induced mice treated with PBS Normal control: Sham treated with PBS Positive control: N.A.	↓ number of pits and percentage of porosity of the skull vs. negative control ↑ BV/TV and BMD vs. negative control ↓ TRAP (+) osteoclasts, RANKL, OSCAR, CTX-1 and ↑ OPG vs. negative control
Kwak et al. [32]	Animals: ICR mice Disease model: LPS-induced bone loss Treatment: 5 μg/g of T-IIA (i.p.) on the day before LPS induction, the day 1, 3, 5 and 7 after LPS induction Negative control: Mice with LPS-induced bone loss Normal control: Mice with PBS treatment Positive control: N.A.	↓ LPS-induced bone loss vs. the negative control ↓ LPS-induced osteoclast formation and loss of cancellous bone vs. the negative control
Zhu et al. [37]	Animals: 64 female Wnt1^sw/sw^ mice with osteoporosis (2.5 months old, 32–40 g) Disease model: Spontaneous *WNT1* mutation for osteogenesis imperfecta Treatment: 10 mg/kg of T-IIA (i.p.)/day for 6 weeks (i.p.) Negative control: Osteoporotic mice with PBS injection Normal control: N.A. Positive control: 10 mg/kg alendronate for 6 weeks	↑ stiffness, ultimate strength, elastic modulus, proline/amide 1, phosphate/amide 1 and phosphate/proline vs. negative control and greater potency than positive control
Zhang et al. [42]	Animals: 40 male C57BL/J6 mice (2 months old) Disease model: STZ-induced diabetic osteoporosis Treatment: 10 and 30 mg/kg of T-IIA 3 times per week for 8 weeks (i.p.) Negative control: Diabetic mice with corn oil (vehicle) Normal control: Non-diabetic mice with corn oil Positive control: 2 mg/kg aliskiren, 3 times a week for 8 weeks	↑ bone mass of trabecular bone vs. negative control ↑ BMD/TV, BV/TV, BA/TA, Conn.D and ↓ SMI vs. negative control
Wang et al. [40]	Animals: 32 female Sprague Dawley rats (3 months old) Disease model: OVX-induced osteoporosis Treatment: 10 mg/kg/day of T-IIA for 2 weeks (i.v.) Negative control: untreated OVX rats Normal control: Sham-operated rats Positive control: N.A.	↑ bone volume, trabecular number, trabecular thickness and ↓ trabecular separation vs. negative control
Cheng et al. [39]	Animals: 18 female C57BL/6 mice (2 months old) Disease model: OVX-induced osteoporosis Treatment: 10 mg/kg T-IIA for 6 weeks (i.p.) Negative control: OVX with normal saline Normal control: Sham with normal saline Positive control: N.A.	↓ trabecular bone loss vs. negative control ↑ BS/TV, BV/TV, BMD, Tb.N and ↓ Tb.Pf vs. negative control
Panwar et al. [30]	Animals: 29 female C57BL/6 mice (8 months old, 25g) Disease model: OVX-induced osteoporosis Treatment: 40 mg/kg/day of T06 for 3 months (oral) Negative control: OVX with an unknown vehicle Normal control: Sham with an unknown vehicle Positive control: N.A.	↑ N.Ob/B.Pm ↓ Tb.Sp and ↑ BV/TV, Tb.N vs. negative control ↑ ALP-positive osteoblasts vs. negative control. ↑ P1NP concentration vs. negative control NS for CatK-positive osteoclast numbers/bone surface and osteoclast numbers/bone perimeter
Cui et al. [43]	Animals: 32 female Sprague-Dawley (4 months old) Disease model: OVX-induced osteoporosis Treatment: 200 mg/kg/day of total tanshinone (5 mg/kg/day of T-IIA and 16 mg/kg/day of C-T) for 10 weeks (oral) Negative control: OVX with oral deionised water Normal control: Sham with oral deionised water Positive control: 30 μg/kg/day of 17α-37 for 10 weeks	↑ BV/TV, Tb.Th and ↓ OCS/BS in LV4 vs. negative control ↑ BV/TV, Tb.Th and ↓ OCS/BS, MAR and BFR/ BV vs. negative control
Zhou et al. [44]	Animals: 30 male and 30 female KM mice (3 months old, 30 ± 5 g) Disease model: retinoic acid-induced osteoporosis Treatment: 40–160 mg/kg/day of tanshinone for 14 days (oral) Negative control: Untreated osteoporotic mice Normal control: sham-operated normal mice Positive control: 3 mg/kg/day of vitamin D_3_ for 14 days	↑ cortical bone thickness and Tb.N with active epiphysis ↑ serum estradiol levels with ↓ serum phosphate, ALP and TRAP levels vs. negative control NS for serum calcium and OCN

Abbreviations: ↑, increase or upregulate; ↓, decrease or downregulate; % ES/BS, percentage eroded surface; % trench surface/BS, percentage trench surface per bone surface; Akt, protein kinase B; ALP, alkaline phosphatase; Apaf-1, apoptotic protease-activating factor 1; BA/TA, bone area fraction; Bcl-2, B-cell lymphoma 2; BFR/BV, bone formation rate per unit of bone volume; BMD, bone mineral density; BMD/TV, BMD over total volume; BMMCs, bone marrow mononuclear cells; BM-MSCs, bone marrow mesenchymal stem cells; BMP, Bone morphogenetic proteins; BS/TV, Bone surface area/total value; BV/TV, bone volume/total volume; BV1/TV, low-density bone volume/Callus total volume; Conn.D, connectivity density; COX2, cyclooxygenase-2; c-Src, proto-oncogene tyrosine-protein kinase Src, C-T, cryptotanshinone; CTR, calcitonin receptor; CTX-1, cross linked C-telopeptide of type I collagen; D-T, 15,16-dihydrotanshinone; ERK, extracellular signal-related kinase; hPDLSC, human periodontal ligament stem cells; IAP, inhibitor of apoptosis protein, IL-1 α, interleukin 1 alpha; IκBα, inhibitor of NF-κB α; IKK-β, IκB kinase-β; iNOS, inducible nitric oxide synthase; i.p., intraperitoneal injection; i.v., intravenous injection; JNK, c-Jun N-terminal kinase; LPS, lipopolysaccharide; LV4, fourth lumbar vertebrae; MAPK, mitogen-activated protein kinase; MAR, mineral apposition rate; M-CSF, macrophage colony-stimulating factor; MMP-9, matrix metalloproteinase 9; N.A., not available; NFATc1, nuclear factor of activated T-cells cytoplasmic 1; NF-κB, nuclear factor kappa B; Nox4, NADPH oxidase 4; NS, not significant; OCN, osteocalcin; OCS/BS, percent osteoclast surface; OPG, osteoprotegerin; OPN, osteopontin; OSCAR, osteoclast associated receptor; Osx, osterix; OVX, ovariectomy; P1NP, procollagen-1 N-terminal peptide; PBS, phosphate buffer saline; PE, polyethylene; PGE2, prostaglandin E2; pNF-κB, phosphorylated NF-κB; PTM, proximal tibial metaphysis; RANKL, receptor activator of nuclear factor κB ligand; RNS, reactive nitrogen species; ROS, reactive oxygen species; Runx2, runt-related transcription factor 2; SMI, structure model index; SOD, superoxide dismutase; STZ, streptozocin; T-I, tanshinone I; T-IIA, tanshinone II A; T06, tanshinone IIA sulfonic sodium; TBARS, thiobarbituric acid reactive substances; Tb.N, trabecular number; Tb.pf, trabecular pattern factor; Tb.Th., trabecular thickness; TMD, tissue mineral density; TNF-α, tumor necrosis factor alpha; TRAP, Tartrate-resistant acid phosphatase; TRAF-1 and 6, Tumor necrosis factor receptor associated factor 1 and 6.

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
