# Peer review of "The Skeletal Effects of Tanshinones: A Review"

_molecules, 2021, doi:10.3390/molecules26082319_

Round 1

Reviewer 1 Report

Comment 1) Summary vs Review article 

The authors described "this systematic review aims to summarise the effects of tanshinones on bone based on evidence from in vitro and in vivo studies."

Potential readers of this journal already knew that review article is totally difrferent from summary of the results. Therefore, the potential readers of this journal would not be interested in this article.

Therefore, summary article should be improved to review article.

Comment 2) Why not shift to the clinical trial ?

The authors enumerated the results of in vitro studies and animal studies.

Effects of tanshinones are examined extensively for long time, many evidences are accumulated in the experimental studies. However, effects of tanshinones are not examined by human trial.

Therefore, the aurhors should discuss the reason why effects of tanshinones are not examined by human trial although they have many evidences.

Comment 3) Effects of specific compound.

In this summary, the authors enumerated the results of specific compounds (tanshinones). This article do not sounds like scientific article because this provides only evidences.

The authors are encouraged to provide the molecular mechanism how tanshiones suppress osteoclast induction so that the potential readers would agree with scientific articles.

Author Response

Thank you for reviewing our manuscript. Your constructive comments are much appreciated and each of them has been addressed in the attached response sheet. 

Reviewer 2 Report

Overall it is good systematic review on a very selected but potentially useful anti-inflammatory and anti-oxidant compound. Authors have extensively reviewed the field and produced a good report on this following standard guidelines.

Author Response

Thank you for the recommendation. 

Reviewer 3 Report

line 10: from Salva to Salvia.

Well the choice of the three database used, well the exclusion operated, well the selection of articles, well and exhaustive performed the review of the 20 articles  and their presentation very useful to readers who,for each article reported, find all the data as the type of the cell or the animal used, the doses, the pathways studied, the type of tanshinone used and the effects of its  treatment, well the division in results from cultured cells studies and from animal studies.Exhaustive the discussion and the references reported.

Author Response

Thank you for the comment and recommendation. 

Reviewer 4 Report

In this systematic review, the authors have discussed the effects of tanshinones, lipid-soluble diterpenoid quinones isolated from Danshen (Salva miltiorrhiza), on osteoblastic bone formation and osteoclastic bone resorption and the possibility of tanshinones as anti-osteoporosis drug.

The process of literature review, article selection, data extraction was well described. The results are well presented in the results section, but it seems there are overlapping areas in the discussion and results.

My concern is about discussion and conclusion. The authors have told that tanshinones can inhibit Cathepsin K (CatK) in osteoclasts, and this effect is selective. However, in table 1, some articles reported tanshinones suppressed MMP-9 expression, actin ring formation, or tartrate-resistant acid phosphatase activity in osteoclasts. I think these findings have indicated that tanshinones are not one of a selective inhibitor of CatK, and this natural compound can generally inhibit the function and differentiation of osteoclasts.

In the part of the discussion, the authors have emphasized that the interaction of osteoclasts and osteoblasts are important for adequate bone metabolism and selective inhibition of Cat K does not influence this. If tanshinones can generally inhibit osteoclastic bone resorption, it is better to reconsider the logic that supports an advantage of tanshinones as a selective inhibitor of CatK.

Minor point

Line 233: please confirm the phrases “cross-linked C-telopeptide of type I collagen (CTX-1) expression in M-CSF and RANKL-treated human and mouse BMCs” Do osteoclasts express type I collagen?

Author Response

Thank you for reviewing our manuscript. Your constructive comments are much appreciated and each of them has been addressed in the attached response sheet. 

This manuscript is a resubmission of an earlier submission. The following is a list of the peer review reports and author responses from that submission.